# PLGA-Lipid Hybrid Nanoparticles for Overcoming Paclitaxel Tolerance in Anoikis-Resistant Lung Cancer Cells

**DOI:** 10.3390/molecules27238295

**Published:** 2022-11-28

**Authors:** Sasivimon Pramual, Kriengsak Lirdprapamongkol, Korakot Atjanasuppat, Papada Chaisuriya, Nuttawee Niamsiri, Jisnuson Svasti

**Affiliations:** 1Laboratory of Biochemistry, Chulabhorn Research Institute, Laksi, Bangkok 10210, Thailand; 2Center of Excellence on Environmental Health and Toxicology (EHT), OPS, MHESI, Thailand; 3Division of Hematology and Oncology, Department of Pediatrics, Faculty of Medicine Ramathibodi Hospital, Mahidol University, Ratchathewi, Bangkok 10400, Thailand; 4Department of Biotechnology, Faculty of Science, Mahidol University, Ratchathewi, Bangkok 10400, Thailand

**Keywords:** PLA-based materials, nanoparticles, drug delivery, paclitaxel resistance, metastasis, lung cancer, anoikis

## Abstract

Drug resistance and metastasis are two major obstacles to cancer chemotherapy. During metastasis, cancer cells can survive as floating cells in the blood or lymphatic circulatory system, due to the acquisition of resistance to anoikis—a programmed cell death activated by loss of extracellular matrix attachment. The anoikis-resistant lung cancer cells also develop drug resistance. In this study, paclitaxel-encapsulated PLGA-lipid hybrid nanoparticles (PLHNPs) were formulated by nanoprecipitation combined with self-assembly. The paclitaxel-PLHNPs had an average particle size of 103.0 ± 1.6 nm and a zeta potential value of −52.9 mV with the monodisperse distribution. Cytotoxicity of the nanoparticles was evaluated in A549 human lung cancer cells cultivated as floating cells under non-adherent conditions, compared with A549 attached cells. The floating cells exhibited anoikis resistance as shown by a lack of caspase-3 activation, in contrast to floating normal epithelial cells. Paclitaxel tolerance was evident in floating cells which had an IC_50_ value of 418.56 nM, compared to an IC_50_ value of 7.88 nM for attached cells. Paclitaxel-PLHNPs significantly reduced the IC_50_ values in both attached cells (IC_50_ value of 0.11 nM, 71.6-fold decrease) and floating cells (IC_50_ value of 1.13 nM, 370.4-fold decrease). This report demonstrated the potential of PLHNPs to improve the efficacy of the chemotherapeutic drug paclitaxel, for eradicating anoikis-resistant lung cancer cells during metastasis.

## 1. Introduction

The two major problems in cancer treatment are drug resistance and the spreading of cancer cells in the body or metastasis. Metastatic tumors are difficult to treat because they frequently develop drug resistance, which remains a major cause of cancer-related death. During metastasis, cancer cells float through the blood and lymphatic circulatory system until they reach targeted locations in distant organs [1]. Normal epithelial and endothelial cells require adhesion to their appropriate extracellular matrix (ECM) and neighboring cells for maintaining their survival. Detachment from ECM or adhesion to inappropriate ECM leads to activation of a programmed cell death termed anoikis in the normal epithelial and endothelial cells [2,3]. This phenomenon avoids the misplaced growth of normal cells in other sites. However, some populations of cancer cells are able to develop anoikis resistance to support their survival when they become floating cells during the journey through the circulatory systems [4]. There is considerable interest in developing approaches for managing anoikis-resistant cancer cells.

Drug resistance of cancer cells in patients can occur before receiving chemotherapy (inherent or intrinsic resistance) or emerge after treatment (acquired resistance), this classification is based on the time when resistance is developed [5]. The effectiveness of current chemotherapeutic drugs is gradually decreased by the acquired resistance after repeated treatments, while the intrinsic resistance of cancer cells causes poor drug response from the first treatment so that patients do not receive the benefit of chemotherapy [6].

New deaths worldwide from lung cancer in 2020 have been estimated to be 1.8 million or 18% of all cancer deaths, making lung cancer the most deadly cancer [7]. A major type of lung cancer is non-small cell lung cancer (NSCLC) which accounts for about 85% of all lung cancer cases [8]. More than 75% of new lung cancer cases were advanced cases at metastatic stage III or IV at the time of diagnosis [9]. Over the past decade, several new treatments for advanced NSCLC have been developed, including targeted therapy and immunotherapy [8]. However, cytotoxic chemotherapy has been the first-line standard regimen for metastatic NSCLC patients who do not have a targetable mutation [10]. Therefore, the improving efficacy of cytotoxic anticancer drugs is still essential for cancer chemotherapy.

Paclitaxel or Taxol^TM^ is a widely used anticancer agent, either as monotherapy or in combination with other drugs, for the treatment of several cancers such as lung, breast, ovarian, prostate, liver, gastric, and bladder cancer [11,12]. Several lines of evidence show that the occurrence of paclitaxel resistance is associated with metastasis in NSCLC patients who have never received paclitaxel treatment. Intrinsic resistance to paclitaxel was observed in 76–79% of the NSCLC patients with metastatic stage III/IV cancer [13,14]. So, we previously used an in vitro non-adherent culture model to mimic floating cancer cells during metastasis and showed that floating H460 lung cancer cells exhibited paclitaxel tolerance that resulted from increased expression of the βIVa-tubulin isotype [15]. Therefore, this model may be used for studying metastasis-associated paclitaxel tolerance in anoikis-resistant lung cancer cells. 

Since paclitaxel-resistant cancer cells require a higher dose of the drug, it is important to explore approaches that enable the increase in intracellular drug concentrations. Paclitaxel is a drug with low solubility in water which requires assistance to allow desired therapeutic concentrations to be reached in tumors: this is a major problem in enabling paclitaxel to achieve satisfactory results [16]. With regards to this issue, a variety of nanoformulation platforms for paclitaxel and other poorly soluble bioactive compounds have been developed to improve solubility and bioavailability [17,18,19]. In addition, nano-sized formulations utilize the concept of enhanced permeability and retention (EPR) effect and passively extravasate through the leaky vasculature of tumor tissues [20].

Different platforms of paclitaxel nanoformulations have been investigated for NSCLC treatment such as albumin-bound nanoparticles [21], solid lipid nanoparticles [22], lipid-based nanoparticles [23], and polymeric micelle nanoparticles [24]. Poly(D,L-lactide-co-glycolide) or PLGA is an FDA-approved copolymer widely used for producing polymeric nanoparticles with biodegradability and biocompatibility. Over the past two decades, PLGA nanoparticles have been used as a carrier of paclitaxel, either as a single drug or in combination with other agents, for lung cancer treatment. Fonseca et al. prepared paclitaxel-loaded PLGA nanoparticles by interfacial deposition method (presently known as a nanoprecipitation method) and showed the enhancement of paclitaxel cytotoxicity in NCI-H69 lung cancer cells, by the nanoparticles compared to free paclitaxel [25]. Recently, Jiménez-López et al. used a modified nanoprecipitation method to prepare paclitaxel-loaded PLGA nanoparticles and demonstrated promising results of the nanoparticles by inhibiting the proliferation of several lung cancer cell lines with an average three-fold reduction in paclitaxel IC_50_ values compared to free drug. Moreover, the paclitaxel nanoparticles also decreased in vitro growth of cancer stem cells and tumor spheroids, as well as resulting in the rapid accumulation of paclitaxel in various tissues including lungs of mice after intravenous administration of paclitaxel nanoparticles, compared with free paclitaxel [26].

Until now, there have been few studies evaluating paclitaxel-loaded PLGA nanoparticles in drug-resistant lung cancer cells. Yuan et al. used PLGA-Tween80 co-polymer to fabricate paclitaxel-loaded PLGA-Tween80 nanoparticles which exhibited a greater effect than paclitaxel-loaded PLGA nanoparticles for facilitating cellular up-take in paclitaxel-resistant A549 lung cancer cells. Furthermore, the IC_50_ value of paclitaxel-PLGA-Tween80 nanoparticles in the resistant cell line was three-fold and eight-fold lower than IC_50_ values of paclitaxel-PLGA nanoparticles and free drug, respectively, indicating that the presence of hydrophilic part of Tween80 on the surface of PLGA-Tween80 nanoparticles effectively improved paclitaxel delivery into the drug-resistant cells [27].

Recently, research trends suggest that PLGA-lipid hybrid nanoparticles (PLHNPs) have a wide range of therapeutic applications. The PLHNPs consist of three parts, (i) an inner PLGA core encapsulating the hydrophobic drug, (ii) a lipid monolayer shell of lecithin coating the PLGA core, and (iii) an outer PEGylated lipid hydrophilic stealth layer of 1,2-distearoyl-sn-glycero-3-phosphoethanolamine-N-[carboxy(polyethylene glycol)-2000] (DSPE-PEG-COOH) interspersed throughout the lecithin monolayer, which prolongs systemic circulation of the nanoparticles by avoiding clearance by the immune system [28]. Previous studies have shown the effectiveness of PLHNP formulations for the treatment of multiple cancers [29]. 

In this work, we aimed to apply PLHNPs to improve the effectiveness of the therapeutic effect of paclitaxel towards anoikis-resistant lung cancer cells during metastasis. Thus, we prepared paclitaxel-PLHNPs and demonstrated their potential for overcoming metastasis-associated paclitaxel tolerance in A549 floating lung cancer cells.

## 2. Results and Discussion

### 2.1. Preparation and Characterization of Paclitaxel-PLHNPs

First, we investigated the effect of solvent used to dissolve the PLGA polymer for the preparation of PLHNPs. Acetonitrile and tetrahydrofuran were chosen as organic solvents in this study. Empty PLHNPs were prepared using a combination of PLGA and mixed lipid through the nanoprecipitation/self-assembly method to form nanoparticles in the nanometer-sized range with a narrow polydispersity index (PDI). As shown in Figure 1a, the particle size of nanoparticles made with PLGA/acetonitrile was smaller than those made with PLGA/tetrahydrofuran. The more water-miscible solvents tended to have greater polarity which resulted in decreasing the size of nanoparticles [30]. Use of acetonitrile as organic solvent compared with tetrahydrofuran resulted in a more highly negative charged surface (Figure 1b). Therefore, acetonitrile has been chosen as an organic solvent for the formulation of paclitaxel-PLHNPs since smaller particle sizes and higher zeta potential were obtained.

The size of formulated paclitaxel-PLHNPs was 103.0 nm with uniform particle distribution (PDI of 0.11). The paclitaxel-PLHNPs possessed a strong negatively charged surface with carboxylic acid end groups of DSPE-PEG-COOH with zeta potential −52.9 mV, indicating a stable nanoparticle dispersion (Table 1). 

To determine the storage stability of nanoparticles, the prepared paclitaxel-PLHNPs were kept at 4 °C and room temperature. As shown in Figure 2, the higher storage temperature induced aggregation of paclitaxel-PLHNPs, causing an increase in particle size. Paclitaxel-PLHNPs were stable in suspension when stored at 4 °C up to 28 days, compared to storage under room temperature. This was consistent with a previous study reporting that PLGA nanospheres should be stored at 4 °C in order to avoid the aggregation [31].

### 2.2. Evaluation of Anoikis Resistance 

In vitro non-adherent culture using polyHEMA-coated plates has been used to obtain floating cancer cells with an anoikis-resistant property that mimics metastasizing cells in the blood and lymphatic circulatory system [15]. Morphological differences were found between A549 attached cells grown as monolayers (Figure 3a) and A549 floating cells cultivated in polyHEMA-coated plates (Figure 3b). The floating cells exhibited a round shape and formed aggregates similar to the aggregated floating lung cancer cells found in lymphatic vessels of lung cancer patients [32].

Next, we studied the activation of caspase-3 which is a hallmark of apoptotic cell death [33]. Anoikis is defined as a type of apoptosis specifically induced by loss of cell attachment, and caspase-3 activation has been used to detect anoikis in cancer cells [32]. We, therefore, used an assay of caspase-3 to evaluate anoikis resistance in our floating cancer cells. Normal epithelial cells undergo anoikis with loss of attachment. Caspase-3 activity of HMECs cultured under non-adherent conditions was significantly increased, compared with the attached cells (Figure 4a), indicating anoikis induction in these cells. In contrast, there was no significant difference between the caspase-3 activities of A549 floating cells and A549 attached cells (Figure 4b). These results demonstrated that anoikis resistance emerged in A549 floating cells when cultured under non-adherent conditions. Taken together, the results confirmed that the A549 floating cells in our in vitro model displayed morphology and anoikis resistance property similar to that found in metastasizing lung cancer cells. Therefore, this model was further used for investigating the effect of paclitaxel-PLHNPs.

### 2.3. Comparison of the Cytotoxic Effect of Paclitaxel-PLHNPs and Free Drug in Anoikis-Resistant A549 Cells

We applied the model of A549 floating cells to evaluate the potential of our formulated paclitaxel-PLHNPs against anoikis-resistant cancer. The A549 attached and floating cells were treated with free paclitaxel and paclitaxel-PLHNPs at concentrations equivalent to 1–1000 nM of paclitaxel for 72 h. As shown in Figure 5, both free paclitaxel and paclitaxel-PLHNPs displayed a similar pattern of dose-dependent cytotoxicity in attached and floating cells. The anoikis-resistant floating cells showed tolerance to free paclitaxel with 53.1-fold resistance, compared with the attached cells (Table 2). Treatment with paclitaxel-PLHNPs could reduce the paclitaxel tolerance of floating cells to a 10.3-fold resistance (Table 2). Our previous studies demonstrated that the acquisition of paclitaxel tolerance in floating lung cancer cells was not due to overexpression of drug transporters such as MDR1/P-gp, but was associated with upregulation of βIVa-tubulin gene, a paclitaxel-resistant β-tubulin isotype [15,34]. 

The remaining 10.3-fold resistance in A549 floating cells after paclitaxel-PLHNPs treatment indicated the contribution of other mechanisms to the paclitaxel tolerance of the floating cells. Several mechanisms can confer paclitaxel resistance in cancer, such as MDR1/P-gp overexpression, altered expression or mutation of β-tubulin isotypes, changes in expression of anti- or pro-apoptotic proteins, and equally important is that each of these mechanisms could separately contribute to the resistance [35]. Since paclitaxel acts by interfering with microtubule dynamics during mitosis in dividing cells, a reduction in cell growth capability may decrease the sensitivity of cells to paclitaxel cytotoxicity. Supporting evidence was reported from gene expression microarray analysis in paclitaxel-resistant MCF-7 breast cancer cells exhibiting slow growth rate, where the cell doubling time and expression of a cell cycle inhibitor gene CDKNA2/p16 were increased in the paclitaxel-resistant cells [36]. Similarly, we observed that our A549 floating cells exhibited a slower growth compared to the attached cells (unpublished data). Prolonging cell division reduces the chance of paclitaxel interfering with microtubule dynamics, making the A549 floating cells have less response to paclitaxel. This resistant mechanism could not be overcome by increasing intracellular drug concentration, and the mechanism might contribute to the remaining 10.3-fold resistance in A549 floating cells after paclitaxel-PLHNPs treatment. 

In A549 attached cells, paclitaxel-PLHNPs showed significantly enhanced cytotoxicity of the drug with a 71.6-fold decrease in IC_50_ value compared to free paclitaxel (Table 2). Interestingly, treatment with paclitaxel-PLHNPs showed a greater cytotoxic effect in A549 floating cells compared with attached cells, as demonstrated by a 370.4-fold decrease in IC_50_ value. These results indicated that the PLHNP platform markedly improved the effectiveness of paclitaxel in anoikis-resistant lung cancer cells. This is in line with a report by Lee et al. [37] demonstrating five-fold higher cytotoxicity of doxorubicin encapsulated in human serum albumin nanoparticles (HSA + DOX NPs), compared with free doxorubicin, in anoikis-resistant MDA-MB-231 breast cancer cells.

The effectiveness of paclitaxel-PLHNPs to overcome paclitaxel tolerance in A549 floating cells might be due to the ability of PLHNPs to deliver the hydrophobic drug into the cells, leading to increased intracellular drug concentration, as proven in our previous study for the use of PLHNPs to deliver a hydrophobic photosensitizer into multidrug-resistant lung cancer cells [34]. 

## 3. Conclusions

This study highlights the importance of paclitaxel delivery using PLHNPs to maximize the efficacy of the drug for overcoming paclitaxel tolerance in metastasizing lung cancer cells. The anoikis resistance property of the A549 floating cells was confirmed by the absence of caspase-3 activation, in contrast to the anoikis induction observed in HMEC floating cells. The self-assembly of PLGA-core and lipid-shell hybrid nanoparticles were formulated through a modified nanoprecipitation technique leading to an average particle diameter of 103.0 nm and strongly negative surface charge with a zeta potential of −52.9 mV. The paclitaxel-PLHNPs showed the ability to increase the therapeutic effect of paclitaxel in A549 floating cells with lower IC_50_ values compared with free paclitaxel. 

Our results suggest that the PLHNP platform has great potential for delivering hydrophobic drugs to treat floating cancer cells during metastasis in the circulatory systems of the body. Further studies in animal models are required for validating the capability of paclitaxel-PLHNPs to reduce metastasis and recurrence.

However, the development of PLHNPs encapsulating other drugs is also necessary, because anoikis-resistant cells of different cancer types might acquire tolerance to distinct drugs. We also suggest that a targeted delivery system or combination with other treatments is required to improve effectiveness for overcoming drug tolerance of anoikis-resistant cancer cells, in order to reduce the metastatic rate and prevent cancer recurrence.

## 4. Materials and Methods

### 4.1. Materials and Chemicals

Poly(D,L-lactide-co-glycolide) (PLGA) with a 50:50 monomer ratio, soybean lecithin consisting of 95% phosphatidylcholine, paclitaxel, 3-(4,5-Dimethyl-2-thiazolyl)-2,5-diphenyl-2H-tetrazolium bromide (MTT), and poly(2-hydroxyethyl methacrylate) or polyHEMA were purchased from Sigma-Aldrich (St. Louis, MO, USA). 1,2-Distearoyl-sn-glycero-3-phosphoethanolamine-N-[carboxy(polyethylene glycol)-2000] (DSPE-PEG-COOH) was obtained from Avanti Polar Lipids, Inc. (Alabaster, AL, USA). Roswell Park Memorial Institute (RPMI) 1640 medium, fetal bovine serum (FBS), and antibiotic–antimycotic solution were supplied by Gibco (Grand Island, NY, USA). All other reagents were of analytical grade and used as received without further purification. Ultrapure water purified by Milli-Q-plus system (Millipore, MA, USA) was used throughout the study.

### 4.2. Nanoparticle Preparation and Characterization

PLGA-lipid hybrid nanoparticles or PLHNPs were prepared through a previously reported nanoprecipitation method combined with self-assembly [38]. In brief, 5 mg of PLGA polymer was dissolved in 2 mL acetonitrile. Lecithin and DSPE-PEG-COOH (3:1, molar ratio) at 20% PLGA weight were dissolved in 10 mL of 4% ethanol. Polymer solution was then slowly dropped into preheated lipid aqueous solution (65 °C) under stirring. The mixture was subsequently stirred at room temperature for 1.5 h. Paclitaxel-PLHNPs were formulated with a similar method where 0.5 mg of paclitaxel was added to 2 mL of PLGA/acetonitrile solution. The resulting nanoparticles were collected and washed three times with water through an Amicon Ultra-15 centrifugal filter, 10 kDa MWCO (Millipore, MA, USA). The nanoparticles were filter-sterilized and stored at 4 °C for later use. 

Hydrodynamic size and size distributions were analyzed by dynamic light scattering (DLS). The zeta potential was determined via electrophoretic mobility. Measurements were performed on samples appropriately diluted with using Zetasizer Nano ZS90 (Malvern Instruments, UK). Paclitaxel content in the nanoparticles was quantified by analyzing the absorbance of paclitaxel in nanoparticles and comparing it to the standard calibration curve of the drug. Nanoparticle stability tests were performed to investigate the effect of storage temperature and duration of storage on the nanoparticle stability in terms of particle size.

### 4.3. Cell Culture

A549 human lung adenocarcinoma cell line was obtained from the American Type Culture Collection (ATCC, Rockville, MD, USA). Normal human mammary epithelial cells (HMECs) were purchased from LONZA (Walkersville, MD, USA). A549 cells were grown as monolayer cultures in RPMI 1640 supplemented with 10% (v/v) FBS and 1% (v/v) antibiotic–antimycotic. HMEC cells were cultured in Mammary Epithelial Cell Growth Medium (MEGM). All cells were maintained at 37 °C in a humidified incubator with 5% CO_2_. 

Floating cells were obtained by cultivating the cells under non-adherent culture conditions using polyHEMA-coated culture plates which were prepared according to a previous report with some modifications [15]. Briefly, 96-well plates were coated with a solution of 30 mg/mL polyHEMA in 95% ethanol, followed by drying at 37 °C for 48 h in a non-CO_2_ incubator for ethanol evaporation. The dried coated plates were sterilized by exposure to UV light for 20 min prior to beginning each test.

### 4.4. Caspase-3 Activity Assay

Anoikis cell death was assessed by determining caspase-3 activation using Caspase-Glo 3/7 Assay (Promega, Medison, WI, USA). Briefly, 1 × 10^4^ cells (100 µL/well) were seeded into non-coated or polyHEMA-coated 96-well plates. After 24 h, Caspase-Glo 3/7 reagent (100 µL) was added to each well. Then, the plates were incubated at room temperature in the dark for 1 h. The resulting luminescence was measured with a luminescence microplate reader (Molecular Devices, Sunnyvale, CA, USA). 

### 4.5. Cytotoxicity Assay

A549 cells were plated into normal 96-well plates at a density of 5 × 10^3^ cells (100 μL/well) to obtain attached cells. The cells were plated into polyHEMA-coated 96-well plates at a density of 1 × 10^4^ cells (100 μL/well) to obtain floating cells. After 24 h incubation, the cells were exposed to different concentrations of paclitaxel or paclitaxel-PLHNPs (25 μL/well) for 72 h. A modified MTT assay for non-adherent culture was employed to determine cell viability at the end of treatment [34]. A 25 µL of fresh culture medium containing MTT was added to each well to reach a final concentration of 0.5 mg/mL and incubated for 4 h, followed by adding 100 µL of lysis solution (20% SDS in 10 mM HCl) to solubilize formazan crystals produced by viable cells, then the plates were kept in the dark for 48 h. After that, absorbance was measured at 550 nm and subtracted from a reference wavelength at 650 nm, using a microplate reader. The cell survival rate was expressed as percentage compared with control. Fold resistance was calculated from ratio of the IC_50_ value in floating cells to the IC_50_ value in attached cells.

### 4.6. Statistical Analysis

Data are expressed as mean with standard deviations (SD) of three independent experiments. A software package PASW Statistics 18 for Windows (SPSS Inc., Chicago, USA) was employed for statistical analysis. A *p*-value less than 0.05 was considered statistically significant.

## Figures and Tables

**Figure 1 molecules-27-08295-f001:**
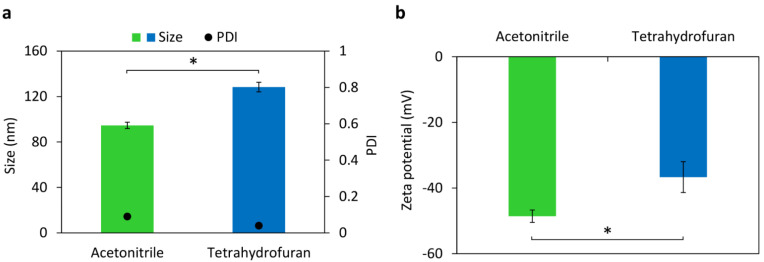
The effect of organic solvents on (**a**) size, PDI, and (**b**) zeta potential of PLGA–lipid hybrid nanoparticles (PLHNPs). Data are reported as mean ± SD from three independent experiments, * *p* < 0.05 compared between the two organic solvents.

**Figure 2 molecules-27-08295-f002:**
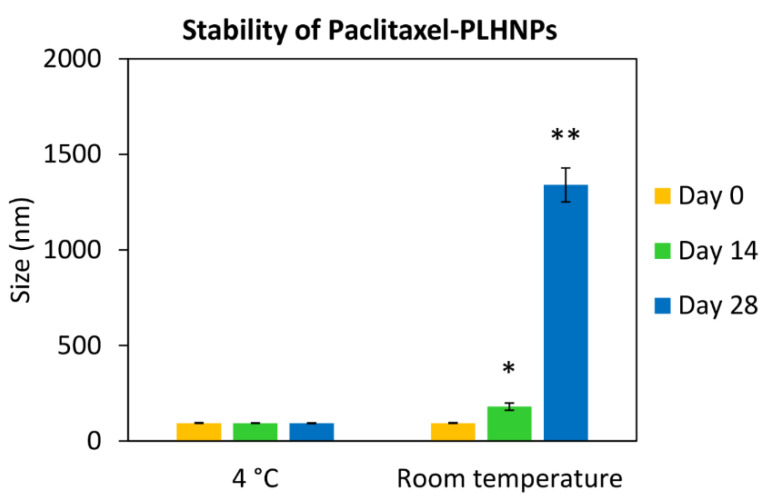
Storage stability of paclitaxel-PLHNPs. Data are reported as mean ± SD from three independent experiments, * and ** *p* < 0.05 compared to Day 0.

**Figure 3 molecules-27-08295-f003:**
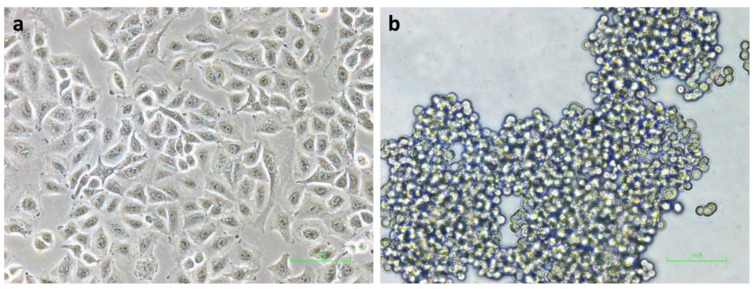
Morphology of (**a**) A549 attached cells cultured under adherent condition and (**b**) A549 floating cells cultured under non-adherent condition. Original magnification of ×200. Scale bar is 100 μm.

**Figure 4 molecules-27-08295-f004:**
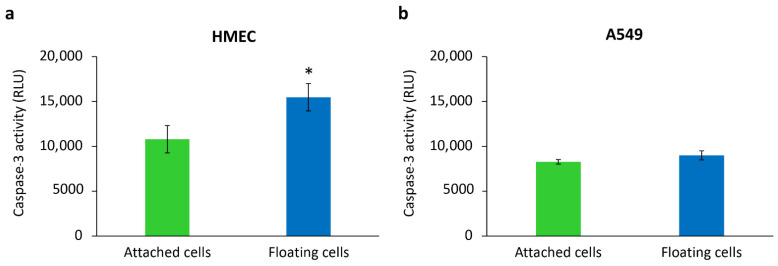
Caspase-3 activation in (**a**) human mammary epithelial cells/HMECs and (**b**) A549 cells after culturing under adherent condition (attached cells) and non-adherent condition (floating cells) for 24 h. Data are reported as mean ± SD from three independent experiments, * *p* < 0.05 compared to the attached cells.

**Figure 5 molecules-27-08295-f005:**
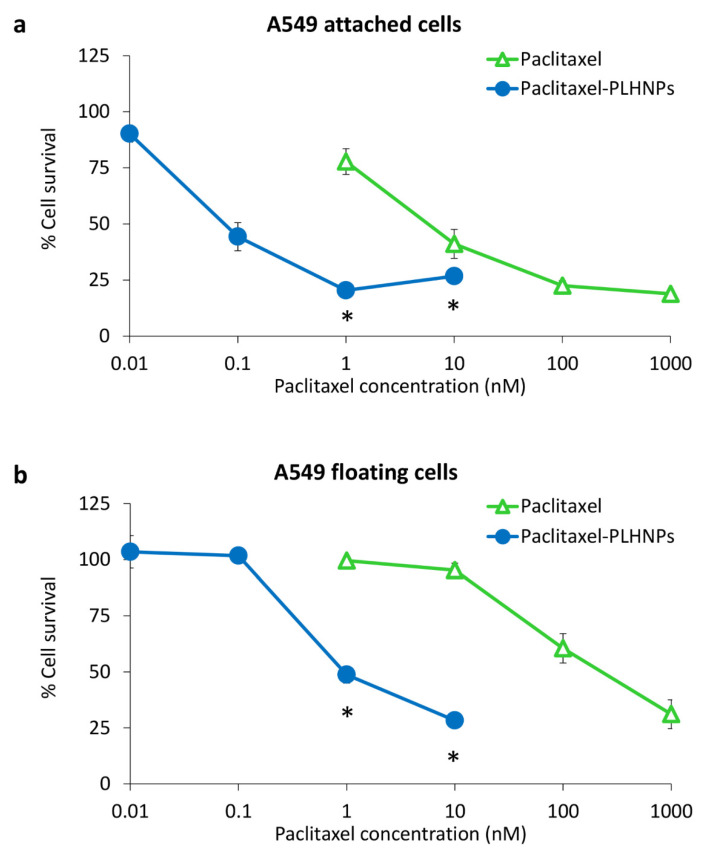
Relative survival rate (%) of (**a**) A549 attached cells and (**b**) A549 floating cells treated with paclitaxel and paclitaxel-PLHNPs. * *p* < 0.05 compared to free paclitaxel.

**Table 1 molecules-27-08295-t001:** Characteristics of paclitaxel-PLHNPs. Data are reported as mean ± SD from three independent experiments. * *p* < 0.05 when comparing empty PLHNPs to paclitaxel-PLHNPs.

	Size	PDI	Zeta Potential
Empty PLHNPs	$*[\begin{array}{c} 94.6\;\pm \;2.7\;nm\\ 103.0\;\pm \;1.6\;nm \end{array}$	0.09	$*[\begin{array}{c} -48.6\;\pm \;1.9\;mV\\ -52.9\;\pm \;2.1\;mV \end{array}$
Paclitaxel-PLHNPs		0.11	

**Table 2 molecules-27-08295-t002:** IC_50_ values of paclitaxel in A549 attached and A549 floating cells. * *p* < 0.05 compared to free paclitaxel.

	IC_50_ at 72 h		Fold Resistance
	A549 Attached Cells	A549 Floating Cells	
Free paclitaxel	$*[\begin{array}{c} 7.88\;\pm \;1.38\;nM\\ 0.11\;\pm \;0.05\;nM \end{array}$	$*[\begin{array}{c} 418.56\;\pm \;194.17\;nM\\ 1.13\;\pm \;0.57\;nM \end{array}$	53.1
Paclitaxel-PLHNPs			10.3
Fold change in IC_50_	71.6-fold decreased	370.4-fold decreased	

## Data Availability

The data presented in this study are available on request from the corresponding author.

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
