# Peer review of "PLGA-Lipid Hybrid Nanoparticles for Overcoming Paclitaxel Tolerance in Anoikis-Resistant Lung Cancer Cells"

_molecules, 2022, doi:10.3390/molecules27238295_

Round 1

Reviewer 1 Report

The research article “PLGA-lipid hybrid nanoparticles for overcoming paclitaxel tolerance in anoikis resistant lung cancer cells” have few corrections that need to be done.

1.      Line 17: Consider drug resistance instead of paclitaxel-specific tolerance.

2.      References were not adequately cited in the article. Consider including relevant references.

Usually, this may give an impression to the readers that authors were not familiar with the research that has already been done, thereby undermining the credibility as an author and the validity of the research.

3.      Line 96: The authors mentioned that “zeta potential -52.9 mV, indicating good stability of dispersed nanoparticles”. How is this justifiable? Theoretically, zeta potentials of more than +30 mV and less than -30 mV are considered as a stable colloidal suspension system that prevents nanoparticle aggregation.

4.      Did the storage stability of nanoparticles carry out as per ICH guidelines? Carried out at 4 °C and room temperature conditions, how about accelerated stability? Both temperature and relative humidity (RH) play a key role in determining the storage stability. But the authors did not considered RH for storage stability.

5.       How the paclitaxel content in the nanoparticles was quantified and what instrument was used for quantification?

Author Response

We appreciate your valuable comments and suggestions which help us to improve quality of the manuscript. We have responded to all of your comments and suggestions. 

Reviewer 2 Report

The work on the “ PLGA-lipid hybrid nanoparticles for overcoming paclitaxel tolerance in anoikis resistant lung cancer cells” is a valuable, and suitable contribution to be published in Molecules Journal after justifying some points.

·        The authors tried in this study to innovate a new nanoparticles of paclitaxel through encapsulate it in PLGA-lipid hybrid nanoparticles (PLHNPs) that make the contribution sounds good. The paclitaxel-PLHNPs had an average particle size of 103.0 ± 1.6 nm, zeta potential value of -52.9 mV with monodisperse distribution and that mean an ideal properties of the nano paricles were achieved. the Cytotoxicity  of the nanoparticles was evaluated in A549 human lung cancer cells by using MTT well-known method. However the revealed results were promising.

·       In the abstract I would like to see the new IC50 values line 25-26

·       in the Keywords it is better to use words rather than phrases or sentences.

·       the Graphical abstract resolution should be improved.

·       The manuscript contain a few number of references and you can improve it with more recent references.

·       the disadvantages of the current anticancer drug like the inherent resistance can be citied with the following recent work https://doi.org/10.1186/s13065-021-00793-8, as well as a recent work for the using of paclitaxel in combination therapy could be mentioned https://doi.org/10.1016/j.molstruc.2022.132345, to improve the MS with more recent ref. rather than the old ones.

·       You can add some new works regarding the using of nano-particles to improve the biological and anticancer activities like, https://doi.org/10.1186/s13065-022-00839-5, https://doi.org/10.1186/s12906-021-03324-z,  as well as I recommended you to write a small paragraph regarding similar methods of innovation of nanoparticles in the improvement of anticancer therapy.

·       Line 76 change word report to work or article.

·       Improve the resolution of figures 1 ,2 and 4

·       Correct the caption of figure 5, the figure is not containing IC50 values but it has % index of the cells

·       why you did not use the same concentrations in the figure 5 for both the drug and and drug nanoparticles ??  

·       The Conclusion should be improved

Best wishes

Author Response

(The authors gave the same response as above.)

Round 2

Reviewer 1 Report

The authors put their efforts into revising the manuscript. Good work.